# Restriction of Glycolysis Increases Serial Killing Capacity of Natural Killer Cells

**DOI:** 10.3390/ijms25052917

**Published:** 2024-03-02

**Authors:** Lea Katharina Picard, Jens Alexander Niemann, Elisabeth Littwitz-Salomon, Herbert Waldmann, Carsten Watzl

**Affiliations:** 1Department for Immunology, Leibniz Research Centre for Working Environment and Human Factors at TU Dortmund (IfADo), D-44139 Dortmund, Germany; lea.picard@uk-koeln.de (L.K.P.); niemann@ifado.de (J.A.N.); 2Institute for Virology, Institute for Translational HIV Research, University Hospital Essen, D-45147 Essen, Germany; elisabeth.littwitz@uni-due.de; 3Max Planck Institute of Molecular Physiology, D-44227 Dortmund, Germany; herbert.waldmann@mpi-dortmund.mpg.de; 4Faculty of Chemistry, Chemical Biology, Technical University Dortmund, D-44227 Dortmund, Germany

**Keywords:** glucose transporter, metabolism, degranulation, cytokines, cytotoxicity

## Abstract

Tumor cells rely heavily on glycolysis to meet their high metabolic demands. While this results in nutrient deprivation within the tumor microenvironment and has negative effects on infiltrating immune cells such as natural killer (NK) cells, it also creates a potential target for cancer therapies. Here we use Glupin, an inhibitor of glucose transporters, to study the effect of limited glucose uptake on NK cells and their anti-tumor functions. Glupin treatment effectively inhibited glucose uptake and restricted glycolysis in NK cells. However, acute treatment had no negative effect on NK cell cytotoxicity or cytokine production. Long-term restriction of glucose uptake via Glupin treatment only delayed NK cell proliferation, as they could switch to glutaminolysis as an alternative energy source. While IFN-γ production was partially impaired, long-term Glupin treatment had no negative effect on degranulation. Interestingly, the serial killing activity of NK cells was even slightly enhanced, possibly due to changes in NAD metabolism. This demonstrates that NK cell cytotoxicity is remarkably robust and insensitive to metabolic disturbances, which makes cellular metabolism an attractive target for immune-mediated tumor therapies.

## 1. Introduction

Tumor cells have a high metabolic demand due to their increased ability to proliferate. Otto Warburg discovered that this metabolic demand is met by anaerobic glycolysis despite the availability of oxygen, in a process known as the Warburg effect [1]. The increased use of anaerobic glycolysis allows tumor cells to produce energy rapidly. However, this also leads to increased production of lactate, resulting in an acidic tumor microenvironment. To prevent cellular acidification, tumor cells upregulate the expression of the lactate transporter MCT4, allowing them to recycle lactate as their main energy source [2]. Because anaerobic glycolysis is an inefficient method of meeting total energy requirements, tumor cells increase their rate of glycolysis 20- to 30-fold compared to healthy cells [3]. This is accompanied by an upregulation of glucose transporters (GLUT) and thus increased glucose uptake. In tumor cells, the expression of GLUT-1 and GLUT-3 in particular is upregulated [2,4]. In addition, increased glucose consumption by the tumor cells leads to nutrient deprivation within the tumor microenvironment (TME). The available energy sources in the TME in turn attenuate the functionality of immune cells such as natural killer (NK) cells [5].

NK cells are lymphocytes belonging to the innate immune system and account for 5–15% of peripheral blood mononuclear cells. They play an important role in immune responses against infections and cancer by killing infected or transformed cells [6]. NK cell cytotoxicity is mediated by activating NK cell receptors such as NKp30, NKp46, NKG2D, 2B4, and CD16 [7]. NK cell effector functions are energy-consuming processes, so cellular metabolism has a major influence on their activity. Two metabolic pathways play important roles in the activation and proliferation of NK cells: oxidative phosphorylation (OxPhos) and glycolysis [8]. Previous studies have shown that NK cells primarily use OxPhos in the resting state, whereas upon activation they increase glycolysis and upregulate OxPhos [5]. This increases energy production and the synthesis of building blocks required for biosynthetic processes that support proliferation and effector molecule production in NK cells [9,10]. Thus, stimulation of NK cells by cytokines such as IL-15 and IL-12 leads to their proliferation and upregulation of OxPhos and glycolysis [5]. In line with this, Keppel et al. have shown that inhibition of OxPhos impairs receptor-mediated IFN-γ production [11]. While glucose is an important source of energy, glutamine and fatty acid metabolism are also important for several cellular functions [12]. Thus, glutamine is known to exert an important influence on NK cell metabolism and effector functions by contributing to the maintenance of the transcription factor cMyc [13]. In contrast, the role of fatty acids in the metabolism of NK cells has not yet been thoroughly investigated.

Given the high glucose dependence of tumor cells, glucose transporters are potential targets for cancer therapy. On one hand, GLUT inhibition could reduce the proliferation of tumor cells, and on the other hand, the TME would have enough glucose for the functional capabilities of immune cells. Based on this, we have recently developed the GLUT inhibitors Glutor [14] and Glupin [15], which are potent inhibitors of glucose uptake. We recently showed that Glutor, an inhibitor of GLUT1, 2, and 3, blocks NK cell proliferation, inhibits IFN-γ secretion, and reduces degranulation [16], which shows that GLUT function is also important for NK cell activities. Here, we were interested in investigating the effects of Glupin, which is a slightly less potent inhibitor of GLUT1 and 3. We treated primary human NK cells with Glupin in short-term and long-term experiments and analyzed their functions, phenotype, and proliferation. Our results show that short-term as well as long-term treatment with Glupin has no negative effect on NK cell activities, except for decreased IFN-γ secretion after long-term treatment. Interestingly, long-term treatment even results in an increase in the serial killing capacity of NK cells.

## 2. Results

### 2.1. Glupin Changes the Energetic Phenotype of NK Cells

First, we tested whether the GLUT inhibitor Glupin exerts an effect on NK cell metabolism. We measured glycolysis and OxPhos in preactivated human NK cells during treatment with different concentrations of Glupin using the Seahorse technology. We found a dose-dependent decrease in ECAR levels upon administration of 100 nM to 200 nM Glupin, whereas OCR levels increased with increasing Glupin concentration (Figure 1). This demonstrates that GLUT inhibition by Glupin reduces glycolysis in NK cells in a dose-dependent manner. Simultaneously, NK cells increase OxPhos to counteract the lack of glucose and to meet energy demands. However, while glycolysis was clearly inhibited, subsequent functional analyses of Glupin-treated NK cells showed no changes in degranulation or IFN-γ secretion after stimulation via CD16 (Appendix A). Further, tumor cell killing was also not affected upon acute Glupin treatment (Appendix A). Thus, consistent with our previous results [16], short-term inhibition of glucose uptake does not impair NK cell effector functions, demonstrating their functional resilience against metabolic disturbances.

### 2.2. Long-Term Treatment with Glupin Altered the Metabolic Profile of NK Cells

To investigate the effect of Glupin on the proliferation of NK cells during long-term treatment, we used freshly isolated NK cells and expanded them with feeder cells in the presence of 100 nM Glupin. This resulted in delayed proliferation within the first 11 days and subsequent normal proliferation in direct comparison to the control, such that no difference in cell numbers was observed after 17 days of culture (Figure 2A). To further investigate this effect of delayed proliferation, we performed flow cytometric analysis of phosphorylated mTOR at days 0, 2, 4, and 7, as pmTOR plays an important role in NK cell proliferation. Glupin treatment resulted in a delayed but sustained and increased phosphorylation of mTOR on day 4 and day 7, respectively. A similar pattern was evident when hexokinase-1 (HK1) was analyzed (Figure 2B), which plays an important role in the onset of glycolysis and is also regulated by mTOR [17]. To determine if Glupin-treated NK cells switch their metabolism to alternative energy sources, the expression of the amino acid transporter CD98 and the transferrin receptor CD71 was analyzed. Long-term treatment with Glupin resulted in significantly increased expression of CD98 as well as CD71 (Figure 2C). However, no differences in cMyc expression levels were detected. To exclude the possibility that NK cells upregulate other GLUT over the course of culture and thus evade Glupin treatment, we performed a glucose uptake assay at day 21. Glupin-treated NK cells showed significantly decreased glucose uptake compared with the control group (Figure 2D), confirming the effect of the inhibitor during the 21-day culture.

To investigate the metabolic changes associated with the inhibition of glucose uptake, we studied energetic metabolism by profiling translational inhibition (SCENITH), a flow cytometric method for analyzing the metabolic profile of cells [18]. This allows conclusions to be made about glucose and mitochondrial dependence, as well as glycolytic capacity and fatty acid oxidation (FAO) in combination with amino acid oxidation (AAO). We observed an increase in mitochondrial dependence and a decrease in glycolytic capacity in Glupin-treated NK cells, whereas glucose dependence and the capacity for fatty acid oxidation and amino acid oxidation were not affected (Figure 2E). These metabolic and expression data confirm that in the absence of glucose uptake, NK cells alter their metabolism to maintain their proliferation capacity.

### 2.3. Glupin-Treatment Increases Serial Killing Capacity of NK Cells

After demonstrating that Glupin affects the metabolism of NK cells but only delays their proliferation, we wanted to take a closer look at the functions of long-term Glupin-treated NK cells. To this end, we first analyzed NK cell activities by stimulating them with plate-bound antibodies against CD16 and then examined degranulation and IFN-γ secretion. We found that CD16-induced IFN-γ secretion was significantly reduced, whereas long-term Glupin-treated NK cells showed no changes in degranulation (Figure 3A,B). A similar pattern was seen after stimulation via NKp30 or via NKG2D + 2B4 (Appendix A). Interestingly, there was no change in IFN-γ production upon co-activation via NKG2D in combination with 2B4 or upon stimulation with IL-12/IL-18, demonstrating that these stimuli can overcome the Glupin-mediated inhibition (Appendix A). The cytotoxic function of NK cells plays a central role in combating cancer cells. While measurement of CD107a detects degranulation of NK cells, we also wanted to directly investigate whether NK cell-mediated killing of tumor cells is affected by Glupin. Therefore, we tested the cytotoxic activity of NK cells treated with Glupin in a ^51^Cr-release assay against K562 target cells. Consistent with the degranulation data, we observed no changes in target cell lysis between Glupin- and control-treated NK cells (Figure 3C). This demonstrates that even though glucose uptake is severely impaired by long-term Glupin treatment, NK cells show a remarkable functional resilience.

Since NK cells are capable of killing not only one tumor cell, but multiple tumor cells in a row, inhibition of glucose uptake might affect the serial killing activity of NK cells. To address this question, we performed a video microscopy-based serial killing assay using long-term Glupin-treated NK cells against MCF7 tumor cells [19], which were previously shown to be resistant to Glupin treatment [15]. Using a 16 h incubation time, we were able to distinguish between cytotoxic and non-cytotoxic contacts for up to six target cell contacts per single NK cell. Interestingly, Glupin-treated NK cells exhibited an increased number of cytotoxic contacts compared to control NK cells (Figure 4A). The number of kills per NK cell, calculated on the basis of these data, showed a slight but significantly increased number of kills per NK cell in the case of Glupin treatment. A similar trend was observed for the percentage of NK cells designated as serial killers (Figure 4B). This indicates that the long-term inhibition of glucose uptake can even have a positive effect on the serial killing capacity of NK cells.

### 2.4. Long-Term Inhibition of Glucose Uptake Results in Increased Levels of Glutamine

To investigate possible mechanisms for this increased serial killing activity of Glupin-treated cells, we first performed a flow-cytometry-based phenotypic analysis. While we detected some changes in the expression of surface receptors on Glupin-treated NK cells (Appendix A), these changes did not explain their increased functional activity. However, we detected significantly higher expression of granzyme B in Glupin-treated cells (Appendix A), while the expression of perforin was unchanged. Next, we performed a metabolomic analysis. As expected, there were reduced levels of glucose and lactate in Glupin-treated NK cells (Appendix A). Interestingly, we found significantly increased levels of glutamine and glutamic acid upon Glupin treatment (Figure 5A), suggesting that NK cells may have increased their glutamine metabolism as an alternative energy source. Therefore, we performed a long-term treatment with the glutaminase inhibitor CB839 in combination with Glupin. The glutaminase inhibitor itself had only a minor effect on NK cell proliferation. However, in combination with Glupin it almost completely blocked proliferation and resulted in cell death after day six of the culture (Figure 5B,C). This demonstrates that glutamine metabolism is essential to maintain cell viability and proliferation under Glupin treatment. To address the question of whether glutamine metabolism is also responsible for the increased serial killing activity of Glupin-treated cells, we first expanded NK cells for 21 days in the presence of Glupin and then treated them with the glutaminase inhibitor CB839 or Glupin individually or in combination. However, this treatment had no clear effect on the functional activity of Glupin-treated cells (Appendix A), suggesting that glutaminase metabolism alone may not be responsible for the increased serial killing activity of Glupin-treated NK cells.

### 2.5. Glupin-Treatment Reduces in NAD+ Metabolism

The metabolism of short-chain fatty acids (SCFA) such as butyrate or pentanoate can increase the cytotoxic activity of T cells [20]. To test this for NK cells, we cultured pre-activated NK cells in the presence of different concentrations of butyrate or pentanoate. This treatment did not significantly change NK cell degranulation after target cell contact (Figure 5D). To exclude other fatty acids, we incubated Glupin-treated NK cells with low-dose Etomoxir (1 μM), an inhibitor of the carnitine palmitoyltransferase 1 (CPT1) (Figure 5E). However, inhibition of CPT1 did not alter the functional capabilities of long-term Glupin-treated NK cells, suggesting that SCFA metabolism is not responsible for the functional activities of Glupin-treated NK cells, which is consistent with the fact that we did not detect any differences in fatty acid oxidation in our SCENITH analysis.

To analyze changes in NK cells treated long-term with Glupin or control in an unbiased manner, we performed bulk RNA sequencing analysis. We identified a total of 149 differentially expressed genes in Glupin-treated compared to control-treated NK cells (Figure 6A). While many of the genes had no immediate relationship to NK cell cytotoxicity, CD38 was among the down-regulated transcripts. Interestingly, we had also detected a reduced CD38 expression in our flow cytometry analysis of Glupin-treated cells (Appendix A). CD38 is an ecto-enzyme which degrades nicotinamide adenine dinucleotide (NAD^+^) into ADP-ribose and nicotinamide. In line with the reduced CD38 expression, we detected significantly increased NAD+/NADH levels in Glupin-treated NK cells (Figure 6C). Therefore, the reduced expression levels of CD38 may be linked to the increased serial killing activity of Glupin-treated NK cells. We additionally performed gene set enrichment analysis and found upregulation of genes related to glycolysis (e.g., MXI1) and hypoxia (e.g., Stannioclacin-2 (STC2) and Liprin-alpha-4 (PPFIA4)), and decreased expression of genes related to oxidative phosphorylation (e.g., ATP6V0E1 and COX11) in Glupin-treated NK cells compared with DMSO-treated controls (Figure 6B). This suggests that Glupin-treated cells attempt to compensate for the lack of glucose uptake by upregulating glycolysis-related genes. However, glucose deficiency also appears to trigger the expression of genes associated with hypoxia, further demonstrating changes in the cellular metabolism of Glupin-treated NK cells.

## 3. Discussion

Here we characterized the effect of the GLUT inhibitor Glupin on the functional activities of human NK cells. Upon the blocking of glucose uptake by Glupin, NK cells immediately showed reduced glycolysis and increased OxPhos, demonstrating their fuel flexibility. However, these acute changes in their metabolism did not affect their functions, as NK cell cytotoxicity and IFN-γ production were not altered upon acute Glupin treatment. This demonstrates that NK cell activities are very robust and are not affected by short-term changes in metabolism and energy production, which is consistent with our previous findings [16]. This could be due to the intracellular storage of glucose and the use of alternative energy sources such as glutamine or fatty acids. Additionally, NK cell cytotoxicity relies mostly on pre-formed effector proteins that are stored within the lytic granules and that just need to be mobilized upon NK cell activation [21]. Therefore, little energy is needed for their immediate effector functions, and it has been described that the low basal metabolic rate of resting NK cells is sufficient to promote acute NK cell effector responses [11].

The stimulation of resting NK cells with feeder cells and cytokines induces NK cell proliferation and activation. This is accompanied by a strong increase in glycolytic activity to provide sufficient energy and precursors for anabolic processes [9,10]. In the presence of Glupin, glucose uptake and therefore glycolysis were severely reduced. Surprisingly, Glupin-treatment only resulted in decreased proliferation for the first 11 days of cultivation. Thereafter, NK cells showed a normal proliferation, suggesting that the Glupin-treated NK cells have to adapt to alternative fuels. Our metabolic profiling found increased concentrations of glutamine and glutamic acid in the Glupin-cultured NK cells. Glutamine is an important fuel for NK cell proliferation and activation [13]. Blocking glutamine metabolism using CB839 in addition to blocking GLUT via Glupin completely blocked NK cell proliferation and resulted in cell death. Therefore, glutamine metabolism is necessary for the continued proliferation and survival of Glupin-treated NK cells. Interestingly, when we previously used the more potent GLUT inhibitor Glutor, NK cell proliferation was completely inhibited [16]. Similarly, NK cell proliferation was much more inhibited in glucose-free medium compared to Glupin treatment (Figure 5C). This suggests that while Glupin severely limits glucose uptake, it does not completely block it. This limited glucose uptake may be necessary to allow the NK cells to switch to alternative energy sources for their proliferation and to retain their functional activity.

Cytokines such as IL-2, IL-15, and IL-21 are known to relay their signals via mTOR, and mTOR plays an important role in the maturation, activation, and metabolism of NK cells [22,23]. Glupin-treated NK cells displayed a delayed increase in the phosphorylation of mTOR in comparison to the control NK cells. This could indicate counter-regulation due to decreased glucose availability. CD98 (SLC3A2) is an amino acid transporter which is upregulated on proliferating and activated lymphocytes [24]. Cytokine stimulation of NK cells leads to an upregulation of their nutrient receptors, such as CD98 and CD71, to fulfil their biosynthetic requirements [25]. Therefore, the increased expression of CD98 and CD71 on Glupin-treated NK cells is a further indication of their metabolic changes in response to a lack of glucose. This is supported by our metabolic SCENITH analysis, which showed that Glupin-treated NK cells had reduced flexibility to use glucose as fuel, and that these NK cells instead strongly depend on mitochondrial OxPhos, showing higher mitochondrial dependency than control NK cells.

Although long term Glupin-treated NK cells had strongly reduced glucose uptake and clear metabolic changes, their degranulation and cytotoxic activity were not impaired. This contrasts with NK cells treated with Glutor, a more potent GLUT inhibitor, which showed reduced degranulation and a block in proliferation [16]. Therefore, the energy obtained via glutamine metabolism and other metabolic adaptations seems not only important for proliferation, but also for the maintenance of NK cell cytotoxic activities. In contrast, IFN-γ production by Glupin-treated NK cells was clearly reduced, demonstrating different metabolic requirements for NK cell cytotoxicity and cytokine production [11].

Interestingly, long-term Glupin-treated NK cells showed increased serial killing activity; we found a slightly higher percentage of serial killers and more kills per NK cell. Our data excluded alternative energy sources such as glutamine or fatty acids as solely responsible for this effect. This is consistent with a previous study, which showed a negative effect of fatty acid metabolism on NK cell function [26]. However, we found increased levels of granzyme B in Glupin-treated NK cells. Additionally, a down-regulation of the NADase CD38 was revealed in our RNAseq and in our flow cytometry analysis. This was consistent with an increase in NAD^+^ levels in Glupin-treated NK cells. Interestingly, antibody-mediated degradation of CD38 was shown to increase NK cell cytotoxic activity [27], and CD38 negative NK cells possess higher metabolic fitness [28]. However, CRISPR/Cas9-mediated deletion of CD38 in primary human NK cells did not result in increased serial killing activity. Therefore, a complete absence of CD38 alone cannot recapitulate our phenotype. In T cells, a reduced expression of CD38 was shown to be associated with higher NAD+, enhanced oxidative phosphorylation, higher glutaminolysis, and altered mitochondrial dynamics that improved their tumor control [29]. This is reminiscent of many of the changes we found in Glupin-treated NK cells and suggests that the reduced CD38 levels, in combination with the increased granzyme B expression and the altered metabolism, may be responsible for their increased serial killing activities.

In summary, reducing glucose uptake via Glupin has only a minor impact on NK cell effector functions and can increase their serial killing capacity while additionally inhibiting the proliferation of tumor cells. This demonstrates that cellular metabolism can be an attractive target for immune-mediated tumor therapies.

## 4. Materials and Methods

### 4.1. Isolation of NK Cells and Cell Culture

Human NK cells were isolated from peripheral blood mononuclear cells of healthy donors with the Dynabeads^®^ Untouched^TM^ Human NK Cell-Kit according to the manufacturer’s instructions (Thermo Fisher Scientific, Darmstadt, Germany). The purity of the NK cells was routinely 95–97% CD56^+^, CD3^−^, 7AAD^−^ as determined by flow cytometry. The study was approved by the Ethics Committee of the Leibniz Research Center (#213) and all blood donors gave informed consent. For experiments with resting NK cells, isolated NK cells were rested in medium overnight and then used for experiments. To generate pre-activated NK cells, isolated NK cells were seeded in 96-well round-bottom plates with irradiated feeder cells (K562-mbIL15-41BBL) with 200 U/mL IL-2 (National Institutes of Health Cytokine Repository, Frederick, MD, USA) and 100 ng/mL IL-21 (Miltenyi Biotec, Bergisch Gladbach, Germany). On day 8, NK cells were re-stimulated with fresh feeder cells and then cultured with 100 U/mL IL-2. On day 14, 2.5 ng/mL recombinant IL-15 (PAN Biotech, Aidenbach, Germany) was added, and after three weeks NK cells were used for experiments. In cases of long-term treatment, Glupin (100 nM) or DMSO (0.1%) were added to the culture every 72 h. For inhibition of CPT1, long-term treated NK cells were treated with 1 µM of the CPT1-inhibitor Etomoxir (Sigma-Aldrich, St. Louis, MO, USA) for 24 h and analyzed for degranulation or IFN-γ secretion as described below. For cultivation with short-chain fatty acids, expanded NK cells were cultivated with 0.1% DMSO, Pentanoate (Sigma-Aldrich; 2.5 mM, 5 mM, 7.5 mM), or Butyrate (Sigma-Aldrich; 0.5 µM, 0.75 µM, 1 µM) for 72 h, followed by a degranulation-assay against K562 targets. K562 cells were cultured in DMEM with 10% FCS and 1% penicillin/streptomycin (all from Thermo Fisher Scientific). MCF7 cells were cultured in DMEM with 10% FCS and 1% penicillin/streptomycin, 1% Sodium Pyruvate, 0.1% NeAA, and 0.1% Insulin (all from Thermo Fisher Scientific). HepG2 cells were cultured in DMEM with 10% FCS and 1% penicillin/streptomycin.

### 4.2. RTCA Analysis

3 × 10^4^ MCF7 cells or HepG2 cells were seeded in e-plates and placed into the xCELLigence device (OMNI Life Science, Bremen, Germany). After 16 h of measurement, 3.75 × 10^3^ NK cells in combination with DMSO (0.1%) or Glupin (100 nM) were added. Cell index was measured every 10 min for 72 h.

### 4.3. ^51^Chromium-Release-Assay

Cytotoxicity was analyzed using a standard 4 h chromium release assay, as already described [30]. In short, target cells were labeled for 1 h at 37 °C and 5% CO_2_ on a Rotator with ^51^Cr (Hartmann Analytic, Braunschweig, Germany). Afterwards, target cells were washed twice and added to the NK cells in a 96 v-well plate. NK cells were washed to remove any residual inhibitor and serially diluted (1:2) starting at an E:T 2:1. After 4 h incubation time, supernatant was collected and analyzed with the Wizard^2^ (PerkinElmer, Waltham, MA, USA) gamma counter. The percentage of specific lysis was determined as follows:$$\frac{experimental\;release-spontaneous\;release}{maximal\;release-spontaneous\;release}\times 100\%$$

### 4.4. Flow Cytometry

To measure degranulation, resting or pre-activated NK cells were treated with Glupin (100 nM) or DMSO (0.1%) for 30 min and then stimulated via plate-bound CD16-mAb (2 µg/mL), NKp30 (2 µg/mL), NKG2D + 2B4 (2 µg/mL), or control IgG (2 µg/mL) in the presence of anti-CD107a PE-Cy5 for 2 h. CD107a-expression was then measured by flow cytometry. For intracellular staining, NK cells were stained for live/dead (Zombie NIR; BioLegend, San Diego, CA, USA) and CD56 using anti-CD56 BV421 (Clone:NCAM16.2; BD Biosciences, San Jose, CA, USA) for 15 min at RT. Intracellular staining was performed using 2% paraformaldehyde for fixation, Permeabilizing Solution 2 (BD Bioscience) for permeabilization and anti-GLUT1 (PE; Clone:202915; R&D Systems, Minneapolis, MN, USA), anti-pmTOR (PE-Cy7; Clone:MRRBY; Thermo Fisher Scientific), and anti-HK1 (AF647; Clone:EPR10134(B); abcam, Cambridge, UK) or Granzyme B (AF647; Clone:GB11; BioLegend) and Perforin (FITC; Clone:dG9; BioLegend). Surface staining was performed using two different panels. Panel 1: Zombie NIR (BioLegend), CD56 (BUV805; Clone:B159; BD Biosciences), CD3 (BUV563; Clone:UCHT1; BD Biosciences), NKp46 (BV421; Clone:9E2; BioLegend), 2B4 (FITC; Clone:C1.7; BioLegend), NKp44 (PerCP-Cy5.5; Clone:P44-8; BioLegend), DNAM-1 (AF647; Clone:DX11; BD Biosciences), NKp30 (APC-Fire750; Clone:P30-15; BioLegend), CD16 (PE-Dazzle; Clone:3G8; BioLegend), and NKG2D (AF700; Clone:FAB139N; R&D Systems). Panel 2: CD38 (BUV395; Clone:HB7; BD Biosciences), NKG2C (BUV496; Clone:134591; BD Biosciences), CD3 (BUV563; Clone:UCHT1; BD Biosciences), KLRB1 (BUV661; Clone:NKR-3G10; BD Biosciences), CD56 (BUV805; Clone:B159; BD Biosciences), 4-1BB (BV421; Clone:4B4-1; BioLegend), KLRG1 (BV510; Clone:2F1; BioLegend), HLA-DR (BV605; Clone:L243; BioLegend), FasL (BV650; Clone:NOK-1, BD Biosciences), TIGIT (BV786; Clone:741182; BD Biosciences), CD11a (AF488; Clone:HI111; BioLegend), CD8 (AF532; Clone:RPA-T8; Thermo Fisher Scientific), CD27 (PerCP-Cy5.5; Clone:M-T271; BD Biosciences), NKG2A (PE; Clone:Z199; Beckmann Coulter, Brea, CA, USA), CTLA-4 (PE-Dazzle; Clone:BNI3; BioLegend), TRAIL (PE-Cy7; Clone:N2B2; BioLegend), PD-1 (AF647; Clone:EH12.1; BD Biosciences), CD18 (AF700; Clone:TS1/18; BioLegend), Tim-3 (APC-Fire 750; Clone:F38-2E2; BioLegend), and Zombie NIR (BioLegend). Cells were measured on a Cytek Aurora spectral flow cytometer and analyzed with FlowJo (Version 10.5.3, FlowJo, LLC, Ashland, OH, USA). Gating strategy: 1. Lymphocytes according to FSC/SSC. 2. Single cells according to FSC-H/FSC-A. 3. Live cells according to Zombie NIR^−^. 4. NK cells according to CD56^+^, CD3^−^.

### 4.5. SCENITH

Assays were performed as described in [16]. In brief, NK cells were incubated in the presence or absence of Oligomycin (1 µM), 2-DG (100 nM) or Oligomycin + 2-DG for 30 min. Next, Puromycin (10 µg/mL) was added for 20 min followed by a wash using ice cold PBS. After an Fc-Block for 5 min, surface staining was performed using Zombie NIR (BioLegend), anti-CD56 BV421 (Clone:NCAM16.2; BD Biosciences), and anti-CD16 PE-Dazzle (Clone:3G8; BioLegend). Cells were then washed, fixed and permeabilized using the Foxp3/Transcription Factor Staining Buffer Set (Thermo Fisher Scientific) followed by intracellular staining using anti-Puromycin AF488 (Clone:12D10; Sigma-Aldrich) and anti-pmTOR PE-Cy7 (Clone:MRRBY; Thermo Fisher Scientific) and analysis using a Cytek Aurora flow cytometer. Glycolytic dependency, mitochondrial dependency, glycolytic capacity, and FAO (fatty acid oxidation) and AAO (amino acid oxidation) capacity were calculated using the formula as already described by Argüello et al. [18].

### 4.6. ELISA

Resting or pre-activated NK cells were treated with Glupin (100 nM) or DMSO (0.1%) for 30 min followed by stimulation via plate-bound CD16-mAb (2 µg/mL), NKp30 (2 µg/mL), NKG2D + 2B4 (2 µg/mL), or control IgG (2 µg/mL), or via IL-12 (0.25 ng/µL) + IL-18 (1.25 ng/µL). as previously described [31]. After 16 h, supernatants were collected and IFN-γ was analyzed using the ELISA MAX™ Deluxe Set Human (BioLegend).

### 4.7. Glucose-Uptake Assay

Glucose uptake-Glo^TM^ Assay (Promega, Fitchburg, WI, USA) was performed according to the manufacturer’s instructions. Luminescence was recorded using a GloMax^R^ instrument (Promega).

### 4.8. NAD/NADH-Assay

NK cells were analyzed for NAD+/NADH concentrations using NAD/NADH-Glo™ Assay (Promega), according to the manufacturer’s instructions. Measurement was done with a GloMax^R^ instrument.

### 4.9. Serial Killing Assays

Assays were performed as described in [19]. In brief, microchips were seeded with MCF7 cells to obtain 60–80 tumor cells/well. After attachment of tumor cells, medium containing the dead cell stain SYTOX Blue (1:1000) was added and the microchip was placed into the incubation chamber of the ApoTome System with an Axio Observer 7 microscope (Zeiss, Jena, Germany) equipped, using a 20×/0.8 Plan-Apochromat objective and an incubation chamber with environmental control (37 °C, 5% CO_2_, humidity device PM S1, PeCon GmbH, Erbach, Germany). NK cells were washed to remove any residual inhibitor and 10–20 NK cells per well were added. Time-lapse microscopy was started. For a duration of 16 h, a picture was taken every 3 min using an Axiocam 506 mono camera. SYTOX Blue was excited using the Colibri 7 LED-module 475 (filter set 38 HE LED) and brightfield was acquired using the TL LED module (both Zeiss, Jena, Germany).

### 4.10. Seahorse Analysis

The energetic phenotype of pre-activated NK cells was analyzed using the glycolysis stress test kit (Agilent, Santa Clara, CA, USA). In brief, culture plates were coated with poly-l-lysine, followed by the addition of pre-activated NK cells. Injection ports were loaded with 10 × concentrated substance (final concentration after injection: Port A: 100 nM Glupin/0.1% DMSO/medium; Port B: 10 mM Glucose; Port C: Oligomycin; Port D: 50 mM 2-DG) and the experiment was performed according to the manufacturer’s instructions (Agilent).

### 4.11. Metabolomics

For metabolomics, 10 × 10^6^ long-term treated NK cells were lysed and analyzed by the Metabolomics Innovation Centre (TMIC Canada) using a combination of direct injection mass spectrometry with a reverse-phase LC-MS/MS custom assay for the targeted identification and quantification of up to 143 different endogenous metabolites including amino acids, acylcarnitines, biogenic amines and derivatives, glycerophospholipids, sphingolipids, and sugars. Isotope-labeled internal standards and other internal standards were used for metabolite quantification. Data analysis was done using Analyst 1.6.2.

### 4.12. RNA-Seq

10 × 10^6^ long-term treated NK cells were used for RNA isolation using an RNAeasy Mini Kit (Qiagen, Venlo, The Netherlands) according to the manufacturer’s instructions, with an additional step of DNAse treatment. RNA-sequencing were performed at the genomics and transcriptomics facility (GTF) at the University Hospital Essen. In brief, the QuantSeq 3′ mRNA-Seq Library Prep Kit FWD was used for library preparation. Sequencing was performed on the Illumina NextSeq500. Reads were processed with standard Illumina base calling and trimmed with Trim Galore v0.6.0 using standard settings. Alignment was performed with hisat2 v2.2.1 to grch38 and standard settings. Raw counts were summed up using Summarize Overlaps from R-Package Genomic Alignments v1.8.4. RNA-Seq data were stored in the Gene expression omnibus archive (accession: GSE216156).

### 4.13. Statistics

Statistical analyses were performed using GraphPad Prism version 9 (GraphPad, La Jolla, CA, USA). The statistical tests used for the analyses are mentioned in the figure legends.

## Figures and Tables

**Figure 1 ijms-25-02917-f001:**
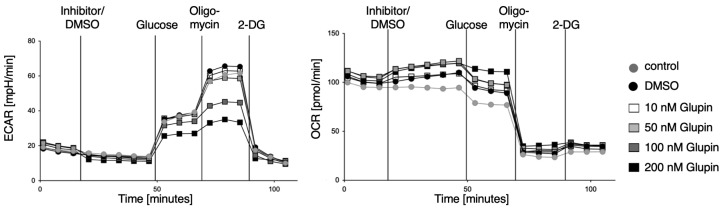
Dose-dependent influence of Glupin on glycolysis in pre-activated NK cells. Representative image of Seahorse analysis. ECAR values and OCR values of pre-activated NK cells before and after injection of Glupin (10 nM, 50 nM, 100 nM, or 200 nM), DMSO, or medium followed by injections of glucose, oligomycin, and 2-DG. *n* = 3.

**Figure 2 ijms-25-02917-f002:**
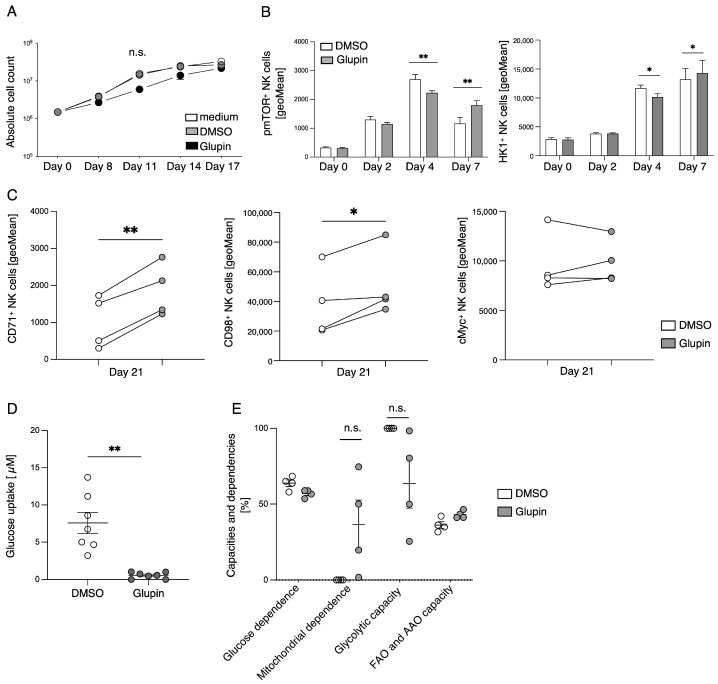
Long-term treatment with Glupin delays proliferation and induces metabolic changes. (**A**) Proliferation of NK cells treated with Glupin compared to control NK cells. Absolute cell numbers were calculated at the indicated time points. *n* = 4. Data were pooled from four independent experiments, each experiment using one donor. (**B**) The expression of HK1 and pmTOR in Glupin-treated and control NK cells was analyzed by flow cytometry at day 0, day 2, day 4, and day 7. *n* = 6. Data were pooled from four independent experiments, with each experiment performed with one or two donors. (**C**) The expression of CD71, CD98 and cMyc on Glupin-treated and control NK cells was analyzed by flow cytometry at day 21. *n* = 4. Data were pooled from four independent experiments, with each experiment performed with one donor. (**D**) Glucose uptake assay of NK cells treated with Glupin compared with control NK cells. *n* = 7. Data were pooled from four independent experiments, with each experiment performed with one or two donors. Mean with SEM. (**E**) The metabolic profile of long-term treated NK cells was analyzed by SCENITH. *n* = 4. Data were pooled from four independent experiments, with each experiment using one donor. Mean with SEM. Statistics: paired *t*-test. Significant differences are indicated by asterisks: * *p* < 0.05, ** *p* < 0.01, n.s., not significant.

**Figure 3 ijms-25-02917-f003:**
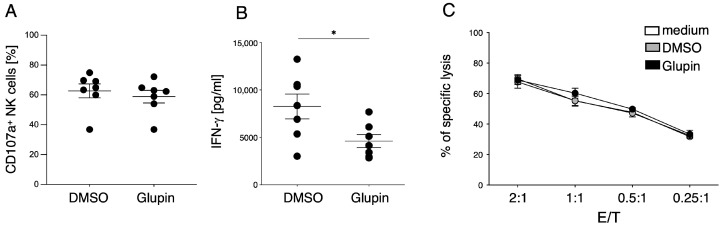
Long-term treatment has no effect on NK cell degranulation. (**A**) Long-term Glupin or DMSO treated NK cells were stimulated with plate-bound antibodies against CD16 for 2 h and then assessed for degranulation. *n* = 7. Data were pooled from seven independent experiments, each experiment using one donor. Mean with SEM. (**B**) Long-treated NK cells were stimulated with plate-bound antibodies against CD16 for 16 h, and IFN-γ secretion was measured by ELISA. *n* = 7. Data were pooled from seven independent experiments, with each experiment performed with one donor. Mean with SEM. (**C**) Specific lysis of K562 by long-term treated NK cells was quantified using the ^51^chromium release assay starting at an E:T of 2:1. *n* = 3. Data were pooled from three experiments, each experiment using one donor. Mean with SEM. Statistics: paired *t*-test. Significant differences are indicated by asterisks: * *p* < 0.05.

**Figure 4 ijms-25-02917-f004:**
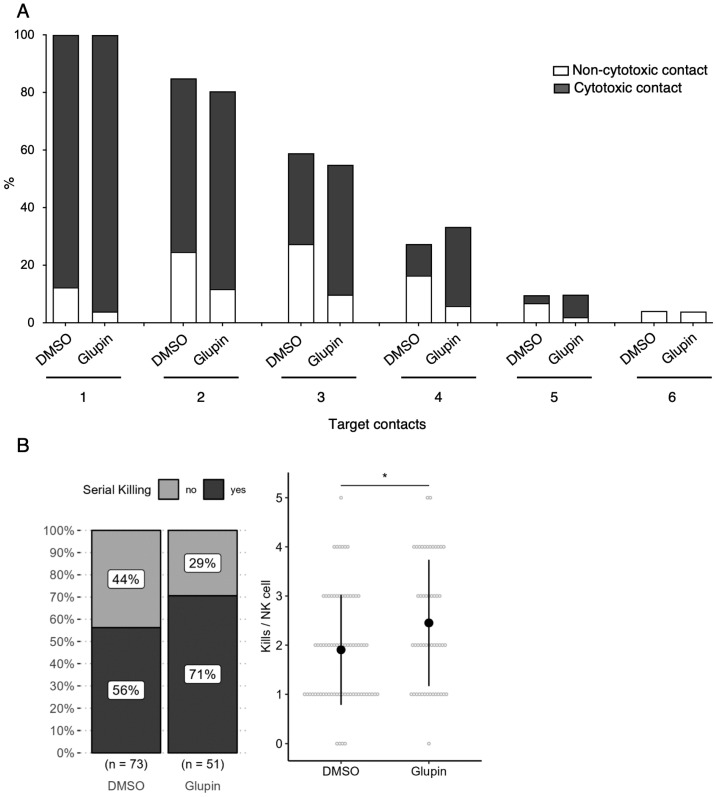
Glupin treatment increases serial killing capacity of NK cells. (**A**) Serial killing assay of long-term Glupin or DMSO treated NK cells as effector cells and MCF7 cells as target cells. The percentage of cytotoxic versus non-cytotoxic contacts was determined for each target cell contact (up to 6 target cell contacts were analyzed). (**B**) Based on these data, the percentage of serial killers (**left** panel) and the kill rate per NK cell (**right** panel) were calculated. The left panel includes the number of NK cells analyzed for each condition from 7 independent experiments, with each experiment using one or two donors. The right panel displays the individual counts of killing events per NK cell as grey circles and the mean values as black dots, with error bars representing the standard deviations for each condition. Statistics: Chi square-test (**left** panel) and Poisson regression for count data (**right** panel). Significant differences are indicated by asterisks * *p* < 0.05.

**Figure 5 ijms-25-02917-f005:**
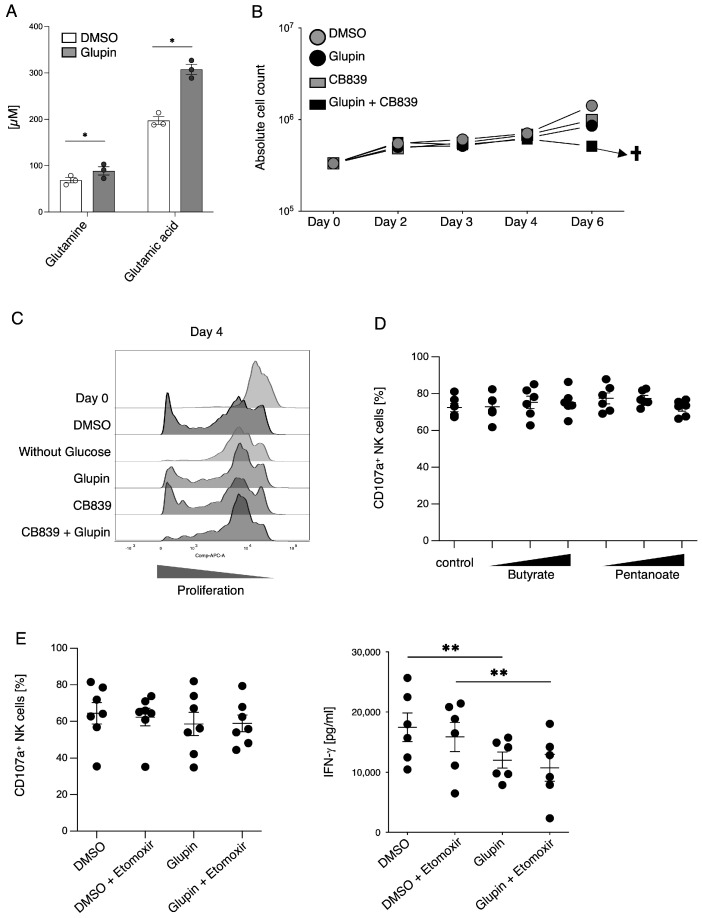
Role of glutamine and fatty acid metabolism in NK cell functions. (**A**) Metabolomics of long-term Glupin or DMSO treated NK cells. *n* = 3. (**B**) Proliferation of Glupin-treated (100 nM), CB839-treated (0.5 µM), or CB839 + Glupin-treated NK cells in comparison to control-treated NK cells. Absolute cell count was calculated at indicated time points. *n* = 4. Data were pooled from three independent experiments; each experiment was performed with one or two donors, †, no more living cells. (**C**) Flow cytometric analysis of proliferation using cell tracker on day 4 of Glupin-treated (100 nM), CB839-treated (0.5 µM) or CB839 + Glupin-treated in comparison to control-treated NK cells and NK cells cultured in media without glucose. (**D**) Preactivated NK cells were incubated with different concentrations of SCFA butyrate (0.5 µM, 0.75 µM, 1 µM) or pentanoate (2.5 mM, 5 mM, 7.5 mM) for 72 h followed by degranulation assay against K562 cells (E:T 2:1). CD107a expression was analyzed by flow cytometry. *n* = 6. Donors were from three independent experiments. Mean value with SEM. (**E**) Long-term treated NK cells (0.1% DMSO or 100 nM Glupin) were treated with or without 1 µM etomoxir for 24 h, followed by plate-bound stimulation via CD16 for 2 h (CD107a assay) or 16 h (IFN-γ). *n* = 6–7. Donors from four independent experiments. Mean with SEM. Statistics: paired *t*-test. Significant differences are indicated by asterisks: * *p* < 0.05, ** *p* < 0.01.

**Figure 6 ijms-25-02917-f006:**
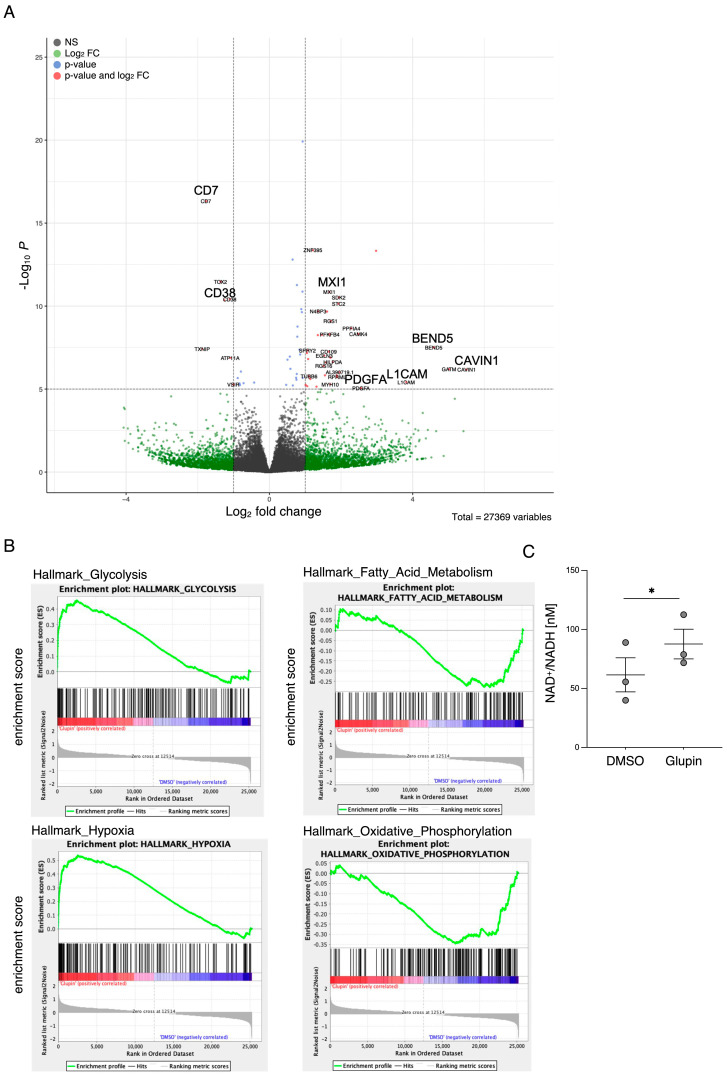
RNA sequencing of Glupin treated NK cells and changes in NAD+ levels. (**A**) Volcano plot demonstrating transcripts differentially regulated in Glupin-treated NK cells compared to control NK cells. Each data point represents a single quantified gene. The y-axis represents the −Log10 *p*-value and the x-axis the Log2 fold change in Glupin-treated NK cells. The threshold for significant genes is chosen for fold change > 1 and for *p*-value cut-off 5. *n* = 6. (**B**) GSEA of long-term treated NK cells. *n* = 6. (**C**) Analysis of NAD+/NADH concentration of NK cells treated with Glupin compared to control NK cells. *n* = 3. Data were pooled from three independent experiments, with each experiment performed with one donor. Statistic: paired *t*-test. Significant differences are indicated by asterisks: * *p* < 0.05.

## Data Availability

The data generated in this study are publicly available in Gene Expression Omnibus (GEO) at GSE216156. Other raw data will be made available upon reasonable request to the corresponding author.

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
