# Peer review of "Restriction of Glycolysis Increases Serial Killing Capacity of Natural Killer Cells"

_ijms, 2024, doi:10.3390/ijms25052917_

Round 1
Reviewer 1 Report
Comments and Suggestions for Authors
In the paper, authors show that the restriction of gluocse uptake by a moderately efficient GLUT inhibitor Glupin leads to stronger effector function of NK cells. This work is an extension of the work published in Cells in 2022, https://doi.org/10.3390/cells11213489 where the same group had reported slightly different results on the same topic. Authors attribute these differences to differences in the effciency of GLUT inhibition by Glutor and Glupin.
This manuscript is well written, data is clearly presented, methods are adequately described and the conclusions are well supported by the reported data. Authors have investigated the mechanism behind the functional benefit to NK cells in depth, which is greatly appreciated.
One comment is, given that Glutor and Glupin showed different results when it comes to the functional changes in the the NK cell phenotype, it would be good to compare them side-by-side in the same serial-killing experiment. This will confirm if there are real differences between Glutor and Glupin treatments and complete the manuscript as this is the only piece missing from the manuscript.
Minor comments:
1. Some sentences are missing references: Line 40-41, line 110-111 etc.
2. Some figures are missing statistics: Fig 1A, Fig 1E. In general, even if the changes are statistically non significant, statistics should be shown (ns in this case) if the observation are scientifically important.
Reviewer 2 Report
Comments and Suggestions for Authors
Pursuit of effective antitumor therapies is of high importance nowadays. The Authors developed and examined if one of the GLUT1 and GLUT 3 inhibitors, namely Glupin, a potential anticancer drug inhibiting glucose intake by tumor cells, has influence on NK cells. They revealed that Glupin in acute treatment only delayed NK cell proliferation while used in long-term treatment had no negative effect on NK cell degranulation but also enhanced serial killing activity of those cells. According to the Authors, cellular metabolism constitutes an attractive target for immune-mediated tumor therapies.
The work is interesting and seems well designed. However, I have some concerns I’d like to express below.
· Keywords. A good practice says that keywords should not repeat the title words
· Please, unify the spelling of “serial-killing” (with or without hyphen)
· Figure legends should be carefully verified in the context of the asterisks. The legend should be consistent with the graphs.
· Describing the chemicals and equipment used in the study, the manufacturer’s name, city (state when appropriate) and country should be provided at first mention.
· The Authors isolated NK cells relaying on CD56 expression. It is known that NK cells can be of phenotype CD16+CD56+, CD16-CD56+, CD16+CD56-. Did you analyze these other subpopulations od NK cells? At least CD16+CD56-. It would be interesting how such cells are affected by treating with Glupin.
· Describing flow cytometric analyses, please provide a gating strategy for the assays used in your experiments.
· In the case of software, the version and manufacturer should be defined.
· Statistical analysis section must be provided. The information about the software used is way too little to assess that the analyses and conclusion drawn from them are correct. I am aware that some information is provided but still, it is too little. The Authors should mention what kind of test were used, how did they verify all assumptions for appropriate tests.
Supplementary Materials section should also be filled in.
